# Cleavage and Polyadenylation-Specific Factor 4 (CPSF4) Expression Is Associated with Enhanced Prostate Cancer Cell Migration and Cell Cycle Dysregulation, In Vitro

**DOI:** 10.3390/ijms241612961

**Published:** 2023-08-19

**Authors:** Muhammad Choudhry, Yaser Gamallat, Sunita Ghosh, Tarek A. Bismar

**Affiliations:** 1Department of Pathology and Laboratory Medicine, Cumming School of Medicine, University of Calgary, Calgary, AB T2N 4N1, Canada; muhammad.choudhry@ucalgary.ca (M.C.); yaser.gamallat@ucalgary.ca (Y.G.); 2Department of Oncology, Biochemistry and Molecular Biology, Cumming School of Medicine, University of Calgary, Calgary, AB T2N 4N1, Canada; 3Department of Medical Oncology, Faculty of Medicine and Dentistry, University of Alberta, Edmonton, AB T6G 2R7, Canada; sunita.ghosh@albertahealthservices.ca; 4Department of Mathematical and Statistical Sciences, University of Alberta, Edmonton, AB T6G 2R7, Canada; 5Tom Baker Cancer Center, Arnie Charbonneau Cancer Institute, Cumming School of Medicine, University of Calgary, Calgary, AB T2N 4N1, Canada

**Keywords:** prostate cancer, CPSF4, Gleason grade, aggressive disease

## Abstract

Potential oncogene cleavage and polyadenylation specific factor 4 (CPSF4) has been linked to several cancer types. However, little research has been conducted on its function in prostate cancer (PCa). In benign, incidental, advanced, and castrate resistant PCa (CRPCa) patient samples, protein expression of CPSF4 was examined on tissue microarray (TMAs) of 353 PCa patients using immunohistochemistry. Using the ‘The Cancer Genome Atlas’ Prostate Adenocarcinoma (TCGA PRAD) database, significant correlations were found between high CPSF4 expression and high-risk genomic abnormalities such as ERG-fusion, ETV1-fusion, and SPOP mutations. Gene Set Enrichment Analysis (GSEA) of CPSF4 revealed evidence for the increase in biological processes such as cellular proliferation and metastasis. We further examined the function of CPSF4 in vitro and confirmed CPSF4 clinical outcomes and its underlying mechanism. Our findings showed a substantial correlation between Gleason groups and CPSF4 protein expression. In vitro, CPSF4 knockdown reduced cell invasion and migration while also causing G1 and G2 arrest in PC3 cell lines. Our findings demonstrate that CPSF4 may be used as a possible biomarker in PCa and support its oncogenic function in cellular proliferation and metastasis.

## 1. Introduction

Prostate cancer (PCa) is the second most common cancer subtype affecting men; it has been found that 1.6 million men worldwide are diagnosed with the disease annually, and has also been shown to cause 366,000 deaths annually [1]. Complex interactions among genetic, environmental, and lifestyle factors play a role in the onset and spread of prostate cancer. The pathology of prostate cancer is characterized by the presence of adenocarcinoma, a type of cancer that arises from the glandular cells of the prostate. PCa is a disease characterized by the aberrant regulation of numerous signaling pathways. In most cases, prostate cancer is an androgen-dependent condition, and the androgen receptor (AR) signaling pathway is critical to its initiation and development [2]. The transcriptional control of genes involved in cell proliferation and survival results from the activation of AR by androgens. However, modifications in AR signaling, such as AR amplification, mutations such as mTOR signaling, or ligand-independent activation, can take place in prostate cancer [3,4]. These alterations aid in the development of the illness and resistance to androgen deprivation therapy. PCa also commonly exhibits dysregulation of the PI3K/AKT/mTOR pathway [5,6]. Constitutive stimulation of AKT and mTOR signaling results from genetic changes including phosphatase and tensin homolog (PTEN) deletion, activating mutations in PI3K, or inactivating mutations in the tumor suppressor gene TSC1/2. This activation encourages cell proliferation, survival, and angiogenesis, which aids in the development of tumors [7,8].

PCa is histologically graded using the Gleason grade grouping system, which is based on the architecture of the glands under a microscope and has been recognized as an improved method for increasing the accuracy of grade stratification [9]. Higher-grade grouping indicates more aggressive cancer and more likely metastatic disease. Other forms of prostate cancer, such as neuroendocrine tumors and sarcomas, can develop in addition to adenocarcinoma, but they are very uncommon. Prostate cancer can develop from a localized tumor to a metastatic malignancy that spreads to bones, lymph nodes, or lungs, among other organ systems. Systemic therapies that target cancer cells throughout the body, such as chemotherapy, hormone therapy, or immunotherapy, are typically used to treat metastatic prostate cancer [10]. However, further advancements in the field are required to enhance treatments against metastatic disease in PCa. Therefore, it is imperative to investigate the molecular mechanisms underlying prostate cancer development and progression, which is essential for the development of new therapies that can improve patient outcomes.

Pre-mRNA is converted into mature mRNA molecules by the vast, dynamic RNA-protein complex known as the spliceosome. Recently, scientific research in the field has uncovered the imperative function of the spliceosome in the development and progression of many different cancer subtypes. Numerous features of cancer genesis and progression have been linked to dysregulation of the spliceosome machinery [11,12]. The impact of the spliceosome on alternative splicing, oncogene activation, tumor suppressor gene inactivation, and therapeutic targeting are all highlighted in this article’s succinct assessment of the state of knowledge about the spliceosome’s role in cancer. Various cancer types have been linked to aberrant alternative splicing events, which produce isoforms with different functionalities [12,13]. These isoforms can encourage the growth, survival, angiogenesis, and metastasis of tumor cells. Alternative splicing can also affect immune evasion strategies and chemotherapeutic response [14]. With the production of spliced variants with increased oncogenic potential, the spliceosome can control how oncogenes are activated. For instance, alternate splicing of the proto-oncogene MYC can produce a more powerful and stable variant, promoting the growth of tumors [15,16]. Exons with crucial tumor suppressor gene sequences can be aberrantly excluded as a result of spliceosome dysregulation. This exclusion may lead to the synthesis of protein isoforms that are truncated or dysfunctional, reducing their ability to control tumor growth and fostering carcinogenesis [15]. The spliceosome has emerged as a possible therapeutic target due to its significance in cancer [17,18]. In preclinical research and early-stage clinical trials, several spliceosome inhibitors have demonstrated promise. These include sudemycins and spliceostatin A [11]. These inhibitors cause cell cycle arrest and apoptosis while preferentially attacking the spliceosome machinery to produce anti-tumor effects [11].

Cleavage and Polyadenylation Specificity Factor Subunit 4 (CPSF4), also known as CPSF30, is a crucial protein involved in the maturation and processing of mRNA in eukaryotic cells [19]. It is important for the cleavage and polyadenylation of pre-mRNA, a process that results in mature mRNA molecules [19]. Within the CPSF complex, CPSF4 performs the role of a scaffold protein by interacting with other subunits to help the pre-mRNA to be recognized and bound to the cleavage site [19]. To generate a stable complex necessary for effective mRNA processing, it has been demonstrated that it interacts with CPSF1, CPSF2, and CPSF3 [19]. To ensure accurate and effective cleavage and polyadenylation, CPSF4 is essential for positioning the cleavage and polyadenylation machinery at the proper place on the pre-mRNA. Additionally, CPSF4 has been linked to the control of alternative polyadenylation, which results in the production of mRNA isoforms with various 3′ untranslated regions (UTRs), which, in HCC, has been shown to depict worse outcomes [20,21].

In conclusion, CPSF4 is an essential part of the CPSF complex, helping to ensure appropriate pre-mRNA cleavage and polyadenylation. Its significance in controlling gene expression and mRNA maturation is highlighted by its function as a scaffold protein and its participation in alternative polyadenylation. The following study aimed to expand the comprehension of mRNA processing and its effects on cellular physiology in the prostate by investigating the precise mechanisms and regulatory roles of CPSF4.

## 2. Results

### 2.1. CPSF4 Expression in Canadian Cohort with Prostate Cancer

Our study’s first objective was to investigate the expression of the CPSF4 protein in our unique TMAs cohort; this group was comprised of patients diagnosed with various prostate cancer stages, such as incidental, advanced, and castrate-resistant PCa (CRPCa). The IHC results showed a successive increase in CPSF4 expression in benign, Gleason group 1 (Gleason score 6 or less) and Gleason Group 2 (Gleason score 3 + 4 = 7) cases. Pathological analysis also revealed CPSF4 expression to be significantly higher in incidental and castrate resistant prostate cancer stages relative to benign prostate tissue (Figure 1A,B). This finding alludes to the possibility that CPSF4 may be essential for the development of prostate cancer, and that the increase in protein expression could potentially correspond to the more enhanced oncogenic capabilities. Overall, these results revealed higher CPSF4 expression in PCa tissue, and further highlight the need for the exploration into the possible diagnostic and prognostic implications of CPSF4 in prostate cancer.

### 2.2. CPSF4 Expression in PAN Cancer Data and TCGA-PRAD Patients Diagnosed with Prostate Cancer

All 22 cancer types had a significantly increased expression of CPSF4 mRNA, according to the Pan-Cancer data analysis (Figure 2A). Particularly in PCa, it was documented that the expression of CPSF4 was significantly higher in tumor tissue as compared to normal tissue (Figure 2B). In comparison to normal tissue and Grade group 1, patients’ tumors with Gleason group 3–5 showed higher CPSF4 expression (Figure 2C). Additionally, we discovered that several genomic aberrations such as *ERG*-fusion, *ETV1*, *ETV4*, and *SPOP*-mutations were substantially associated with overexpression of CPSF4 (Figure 2D).

### 2.3. CPSF4 Gene Set Enrichment Analysis from TCGA-PRAD Database

To further investigate the related gene sets associated with *CPSF4* from the TCGA PRAD cohort, we next performed a gene set enrichment analysis (GSEA). The 50 most significantly associated genes in each subset were shown, along with two heat maps that we created to show the positively and negatively correlated genes (Figure 3A). With the help of plasma membrane adhesion molecules, cell adhesion via plasma membrane adhesion molecules, cell junction organization, and other biological processes were all downregulated, according to our study of the GSEA (Figure 3B). Additionally, we discovered that there was a significantly lower expression of genes and pathways related to the membrane area and protein complex involved in cell adhesion (FDR 0.05) (Figure 3B). The heat maps of the positively and negatively linked proteins (Figure 3C) then demonstrate how the expression of CPSF4 RNA was correlated to the proteome. This revealed to us that CPSF4 expression was shown to have a positive correlation with well-known oncogenes such as Cyclin B1, FOXM1, Chk1/2, and c-Myc (Figure 3C). In Figure 3C a negative correlation can be seen between CPSF4 and beta-catenin, ECAD, MAPK1, and RICTOR, all of which have been classified as known tumor suppressors. These findings prompted further in vitro studies by implying that CPSF4 may be an important regulator of various important biological, physiological, and molecular processes involved in prostate cancer cell-line proliferation and metastatic capacity. This may suggest that CPSF4 could offer insightful information about disease progression and could potentially be investigated as a marker of diagnostic and/or therapeutic importance.

### 2.4. CPSF4 Expression in PCa Cell Lines and Association with Current Markers

Western blot was used to assess the levels of CPSF4 protein expression in the HEK-293, RWP-1, DU-145, PC3, and PC3-ERG cell lines (Figure 4A). On the cell lines DU-145, PC3, and PC3-ERG, CPSF4 knockdown was accomplished successfully. The best time for the knockdown to occur was 48 h after treatment, according to a Western blot examination of CPSF4 expression (Figure 4B). CPSF4 siRNA #2 was shown to have a better knockdown efficiency. Additionally, we noticed a considerable downregulation of vimentin in DU-145 cell lines after CPSF4 knockdown, which was assessed as a 71.8% decrease compared to the NC control (Figure 4C,D). In PC3 cell lines, E-Cadherin expression decreased by 29.1%, but not significantly in DU-145 cells (Figure 4C,D). When looking at c-Myc, both DU-145 and PC3 cells were shown to exhibit a decrease in expression with a 37.2% and 73.3% reduction, respectively (Figure 4C,D). Moreover, β-catenin showed opposing effects, as it exhibited a 57.5% decrease in DU-145 and a 105.3% increase in PC3 cells, when treated with siCPSF4 (Figure 4C,D).

### 2.5. Knockdown of CPSF4 Attenuates the Migration and Invasion of PCa Cells, In Vitro

Understanding the function of CPSF4 in the process of metastasis is a crucial aspect of the progression of cancer. Additional tests were carried out to examine the impact of CPSF4 inhibition on the metastatic phenotype of PCa cells. Migration and invasion in vitro assay were carried out to evaluate whether the inhibition of CPSF4 would be associated with higher or lower Pca cells’ capacity to migrate and invade to clarify whether CPSF4 plays a role in causing a metastatic phenotype. An analysis of the invasion assay findings revealed that both PC3 and DU-145 cells’ ability to invade was decreased because of CPSF4 downregulation (Figure 5A). Treatment with siCPSF4 was quantified and showed a decreased invasive cell count by 67.2% and 53.4% in DU-145 and PC3 cells, respectively (Figure 5B). Additionally, the migration assay showed that CPSF4 downregulation decreased the capacity of both PC3 and DU-145 cells to migrate (Figure 5C). In DU-145 and PC3 cells, siCPSF4 knockdown was shown to reduce the number of migratory cells by 64% and 60.4%, respectively (Figure 5D).

### 2.6. In Vitro Effects of CPSF4 Expression on PCa Cell Cycle Progression and Proliferation

Using flowcytometry, cell cycle assay was performed using PI. to investigate the function of CPSF4 as a cell cycle regulator. The results in DU-145 cells revealed that 48 h after CPSF4 was knocked down, the number of G1 cells increased by 11%, whereas the numbers of cells in G2 and S phases decreased by 2% and 8%, respectively (*p* < 0.01) (Figure 6A). In PC3 cells, the populations of G1 and S phase cells decreased by 3% and 12%, respectively, while the populations of G2 cells increased by 14% (*p* < 0.01) (Figure 6B). In terms of apoptotic cell populations, neither cell line significantly differed from the other (Figure 6A,B). The results from the colony formation assay depicted decreased cell survival and reproduction in siCPSF4-treated cells; this was shown by 60.1% and 75.6% decreases in the cell number for DU-145 and PC3 cells, respectively (Figure 6C,D).

## 3. Discussion

In this study, it was demonstrated in our cohort that higher levels of CPSF4 protein expression was associated with aggressive PCa subtypes, with the highest levels of CPSF4 intensity being identified in PCa patients diagnosed with incidental and castrate resistant PCa. RNAseq data analysis from TCGA-PRAD database confirmed that high CPSF4 expression was not a feature specific to prostate cancer, but instead was a common characteristic of the 22 cancer subtypes analyzed within this study. The TCGA-PRAD database also confirmed that Gleason grade grouping correlated positively with higher RNA CPSF4 expression. Additionally, a positive correlation between CPSF4 expression and the presence of common genetic aberrations, including ERG-fusion, ETV1-fusion, and SPOP-mutations, were observed to be significant within the cohort. No previous literature has documented the relationship of these mutations with CPSF4 and, so, this could be of interest for future studies.

Gene set enrichment analysis revealed cell adhesion, motility, and membrane composition were shown to be downregulated in correlation with CPSF4 expression. This demonstrated dysregulation of pathways that were important in controlling cell migration. These findings implicated the potential role of CPSF4 and its involvement in cell invasion, migration, and cytoskeletal dynamics, which may contribute to a more metastatic phenotype in cancer cells. This was later confirmed in the study by performing migration and invasion confirming the role of CPSF4 as an oncogenic of metastatic processes. CPSF4 RNA expression was also used to generate heatmaps to depict the positively and negatively correlated proteins in patients from the TCGA-PRAD database. From this, we observed numerous cell cycle regulating proteins showing a strong correlation, such as Chk1/2, c-Myc, and RICTOR, to name a few. These proteins have been previously implicated as regulators of cancer cell cycle progression [22,23,24,25]. Furthermore, our findings along with those in the field implicate CPSF4 involved as being a crucial regulator of cell cycle progression and its association with both crucial tumor suppressors and oncogenes implicates it as a potential multilevel regulator of the proliferation process.

Our results showed that in vitro CPSF4 knockdown was associated with reduction in migration and invasion of PCa cells. Cells inhibited for CPSF4 expression showed more than a 50% reduction in their metastatic capabilities. Additionally, the Western blot experiments showed that CPSF4 knockdown decreased Vimentin expression in DU-145 and raised ECAD expression in PC3 cells, supporting the possibility of potential mechanism through which CPSF4 enacts its oncogenic function as a metastatic marker. These results are in agreement with prior studies, in triple negative breast cancer (TNBC) documenting enhance metastasis in vivo and in vitro [26]. These metastatic effects were also observed in lung, colon, and oral squamous cell carcinoma (OSCC) [27,28,29,30]. CPSF4 being involved in migration and invasion implicates it as an oncogenic driver of metastasis that could potentially be targeted to prevent progression and future recurrence of the disease.

Our in vitro research showed that CPSF4 is linked to abnormal proliferation of PCa cells. In PC3 cells, there was a 14% increase in G2 cell population, which could potentially be indicative of G2/M cell cycle arrest. The inhibition of CPSF complex (CPSF30, 160, 73, and 100) has previously been shown to increase G2/M arrest in MDA-MB-231 TNBC cells [31,32]. However, in DU-145 cells, G1 arrest was observed with an 11% increase in the cell population; this could be explained by the decrease in c-Myc and β-catenin observed in the Western blot analysis. This is supported by a study in OSCC, where it was found that a reduction in CPSF4 led to G1 arrest through a c-Myc-targeting mechanism [28]. Additionally, Wnt/β-catenin signaling has also been shown to drive c-Myc driven proliferation in teratocarcinoma cells in vivo [33]. Increased G1 cell population was also observed in hepatocellular carcinoma when the CPSF complex was inhibited [21]. The colony formation results also are supported by the study on OSCC that saw a reduction in cell viability and proliferation when treated with siCPSF4, in vivo [28]. These results, along with findings in the research field, confirm that CPSF4 is involved in the regulation and progression of cells through the cell cycle, as inhibition was shown to lead to cell cycle arrest at different stages in different PCa cell types. This is further supported by the drastic decrease in colony formation showing decreased replication and proliferative capacity. It is plausible for CPSF4 to decrease proliferation of PCa cells by decreasing replicative capacity induced by the reduction in aberrant proteins resulting from errors in splicing. The inhibition of CPSF4 may also work to reduce total protein content of the cell by triggering mRNA instability through inhibition of polyadenylation, potentially leading to increased degradation of pre-mRNA transcripts and, thus, decreased translation of these proteins. Further studies are needed to document the mechanisms underpinning CPSF4 effects on the proliferation pathways in PCa and other forms of cancer.

## 4. Materials and Methods

### 4.1. Study Population and Pathological Analysis

A cohort of 353 patients diagnosed with incidental, advanced, and castrate resistant PCa were investigated for this study. Patients in this study all provided written informed consent and the use of these samples was approved by the Ethics Committee. Tumor tissue was collected via transurethral resection of the prostate (TURP) and placed on a tissue micro-array (TMA) of the two predominant Gleason scores. CPSF4 expression was analyzed using a three-tiered scoring system to categorize the intensity of CPSF4 levels (0, negative; 1, weak; 2, moderate; and 3, strong). The study pathologist (TAB) assigned Gleason grades according on the 2018 WHO and ISUP grade group.

### 4.2. Immunohistochemistry (IHC)

IHC labelling was used to measure the expression of the protein CPSF4. Formalin-fixed paraffin-embedded (FFPE) slices with a 4 μm thickness were placed on microscope slides and were stained with the Dako Omnis Auto Stainer, following the standard recommended procedure (DAKO Omnis Stainer User Manual (2019) (Agilent Technologies, Santa Clara, CA, USA). Briefly, tissue sections that had been deparaffinized and rehydrated with ethanol at progressively lower concentrations. Epitope retrievals were performed in Tris-buffer (pH 9.0) was used for the antigen retrieval process. The CPSF4 antibody utilized was rabbit monoclonal Cat# 70997 (1:50) from Cell Signaling in (Cell Signaling, Danvers, MA, USA). The FLEX DAB+ Substrate Chromogen system Cat# GV82511-2 (Agilent Technologies, Santa Clara, CA, USA) was the detection reagent utilized.

### 4.3. TCGA PRAD Data Analysis

This cohort included 497 male patients (*n* = 497) to examine the expression of CPSF4 from the prostate adenocarcinoma (TCGA PRAD) database found in The Cancer Genome Atlas (TCGA) database. The RNAseq database that was taken from TCGA was examined by the bioinformatic analysis using UALCAN (https://ualcan.path.uab.edu/analysis.html accessed on 17 March 2023) and LinkedOmics (https://www.linkedomics.org/login.php accessed on 4 April 2023) [34,35]. This online tool compares RNAseq data for a particular gene of interest using quick analysis servers. Additionally, we investigated the RNA expression of the CPSF4 in normal and tumor tissue from the TCGA PRAD database for expression that is exclusive to prostate cancer. Welch’s *t*-test calculated the significance of variations in expression levels between normal and main tumors or tumor subtypes.

We used LinkedOmics (http://www.linkedomics.org (accessed on 17 February 2023)) to create a Gene Set Enrichment Analysis (GSEA) from the TCGA PRAD database in order to investigate the biological, cellular, and molecular effects of aberrant CPSF4 expression. This investigation utilized a web-based toolbox and explorer and was based on FDR.

### 4.4. Cell Lines

The study’s cell lines for human prostate cancer included HEK293, RWP-1, PC3, and DU145 cells, which were obtained from The American Type Culture Collection (ATCC; Manassas, CA, USA). The University of Michigan’s Felix Feng provided stable PC3-ERG cell lines.

Prostate cancer cells PC3 and PC3-ERG were grown in DMEM/F12 (GIBCO life technology, Grand Island, NY, USA). In DMEM media (GIBCO life technology, Grand Island, NY, USA), HEK293 and DU145 cell lines were grown. RWPE-1 cells were grown in keratinocyte-free serum medium (K-FSM) media (GIBCO life technology, Grand Island, NY, USA). Additionally, 10% FBS (GIBCO life technology, Grand Island, NY, USA) and 1% Pen Strp (penicillin/streptomycin (Ref# 15140-122, GIBCO life technology, Grand Island, NY, USA) were added to all the media previously mentioned. Atmospheres of 5% CO_2_ and 37 °C were used to incubate the cells.

### 4.5. Cell Line Transfection and RNA Silencing

The CPSF4 gene was knocked down using short interfering RNAs (Ambion, Grand Island, NY, USA), which provided a pre-designed siRNA silencer and a scrambled siRNA (as a negative control). Briefly, PC3 and DU-145 cells were seeded on six-well plates and grown there until they were 70–80% confluent. Next, the Lipofectamine RNAiMAX (Invitrogen, Carlsbad, CA, USA) and Opti-MEM (GIBCO life technology, Grand Island, NY, USA) reaction mix for siRNA transfection was made in accordance with the manufacturer’s instructions. The effectiveness and duration of the temporary CPSF4 knockdown were evaluated using a Western blot analysis to confirm the knockdown of CPSF4.

### 4.6. Western Blot

Total protein was extracted using RIPA lysis buffer (Cat #9806, Cell Signaling, Danvers, MA, USA) with the addition of 1:100 protease PMSF inhibitors (Cat # 5872S, Cell Signaling, Danvers, MA, USA). Equal amounts of proteins were injected into each well and separated using SDS-PAGE gel electrophoresis before the Western blot was conducted. The proteins were then transferred to the PVDF membrane BIO-RAD Immuno-Blot^®^ Membrane. Nonspecific binding was stopped once the transfer was finished by shaking the membrane in a blocking buffer made of 10% skimmed milk in Tris-Buffer solution pH 7.4 (TBS) for one hour at room temperature. The membranes were next exposed to a primary antibody for an overnight incubation at 4 °C. Then, for 1 h at 37 °C, either an anti-rabbit IgG or anti-mouse IgG secondary antibody conjugated to HRP horseradish peroxidase (Cell Signaling, Danvers, MA, USA) was added. After primary and secondary antibody incubations, washing processes were carried out. These required three washes with TBS buffer + 1% Tween for 5 min each. The membraned was visualized using ECL substrate Chemiluminescence signal on Bio-Rad Laboratories ChemiDoc imaging equipment (Hercules, CA, USA).

### 4.7. Migration and Invasion Assay

As previously mentioned, knockdown was performed on the PC3 and DU-145 cell lines. After being transfected for twenty-four hours, the cells were trypsinated before being moved to either a Corning Matrigel invasion chamber (Ref# 354,480, Corning, Bedford, MA, USA) for the invasion assay or the top of a Corning BioCoat control insert for the migration assay (Ref# 354,578, Corning, Bedford, MA, USA). The cells were fixed and stained with Diff Quick (Siemens Healthcare Diagnostics, Tarrytown, NY, USA) after 48 h had passed. Images were then developed using an inverted EVOS FL Life microscope, and bright field images at 10 and 40 magnifications were taken. Utilizing several frames that were counted, averaged from a magnification of 40, and compared to the negative control for analysis, the number of cells was determined.

### 4.8. Flow Cytometry

We carried out a flowcytometry cell cycle investigation to examine the impact of CPSF4 on these processes. In a nutshell, the preparation for the knockdown of CPSF4 and the matching control groups with the requisite number of replications was conducted as previously stated. The cells were harvested, rinsed in cold PBS, fixed in 70% ethanol, and stained using FxCycleTM PI/RNase Staining Solution from Invitrogen in Carlsbad, California, USA, which contained 50 g/mL propidium iodide (PI) and 100 g/mL RNase A. Using a BD LSR II Flow Cytometer, the DNA composition of the cells was examined. With the help of FlowJo^TM^ v.10 Software-BD Biosciences (Franklin Lakes, NJ, USA), the results were further examined and exported as graphics.

### 4.9. Colony Formation Assay

Following CPSF4 siRNA transfection, cell viability was assessed by colony formation assay. DU-145 and PC3 cells were collected, counted, and plated in triplicates into 6-well plates (400 cells/well) 48 h after transfection. The cells were cultured in medium containing 1% FBS + 1% penicillin/streptomycin. Two weeks after the cells were plated, the colonies were fixed, stained, and counted.

### 4.10. Statistical Analysis

The statistical analysis was performed, and graphs were plotted using GraphPad Prism (v 9.1.0). To compare the two groups, the unpaired *t*-test was used. *p* values less than 0.05 were regarded as significant. All values were presented as Mean ± SEM or Mean ± SD.

## 5. Conclusions

In conclusion, our research supports CPSF4′s oncogenic function in PCa. We established that CPSF4 is overexpressed and linked to an accelerated clinical stage in several malignancies, including PCa. Furthermore, greater Gleason grade groups and patients’ clinical prognosis were related to CPSF4 overexpression. Additionally, CPSF4 knockdown was found to cause G1/S or G2/M arrest while lowering PCa cell invasion and migration potential. Our findings showed that CPSF4 operates as a cell cycle and proliferative regulator in connection with c-Myc.

## Figures and Tables

**Figure 1 ijms-24-12961-f001:**
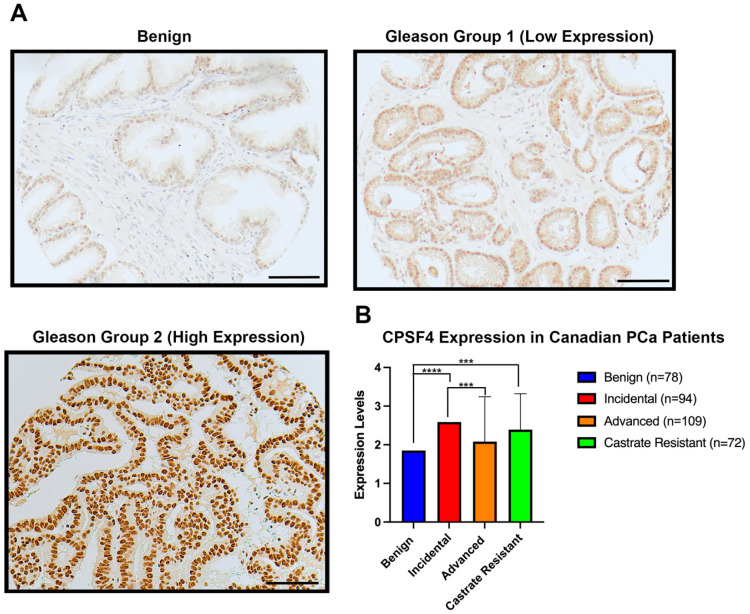
Expression of CPSF4 in a clinical cohort of prostate cancer patients. (**A**) Immunohistochemistry-staining images of CPSF4 expression in Gleason grouping. Images were viewed at 20 times magnification (scale bar: 100 um). (**B**) Bar plots depict the expression of CPSF4 in benign, incidental, advanced, and CRPC clinicopathological subtypes of prostate cancer (*n* = 78, 94, 109, and 72, respectively). Student’s *t*-test was used to determine the significance of the data, which are shown together with Mean ± SD bars. (Asterisk *** indicates *p* < 0.001 and **** indicates *p* < 0.0001).

**Figure 2 ijms-24-12961-f002:**
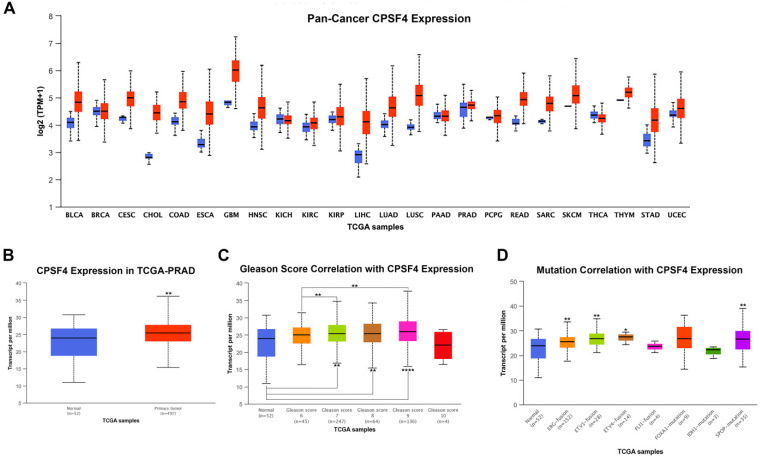
*CPSF4* gene expression study using RNAseq in normal and tumor tissues. (**A**) Boxplots show the expression of the *CPSF4* gene in tumors (red) and normal (blue) in 22 different forms of cancer. (**B**) Boxplot showing the expression of the *CPSF4* gene by RNA-seq in normal tissue (blue) and PCa tumor (red) tissue. (**C**) Correlation of the expression of *CPSF4* RNA with patient Gleason scores. (**D**) Boxplots demonstrating the relationship between *CPSF4* and common mutations in PCa (*ERG, ETV1, ETV4, FLI1, FOXA1, IDH1,* and *SPOP*). (Asterisks denote significant *p*-values; * *p* < 0.05, ** *p* < 0.01 and **** indicates *p* < 0.0001).

**Figure 3 ijms-24-12961-f003:**
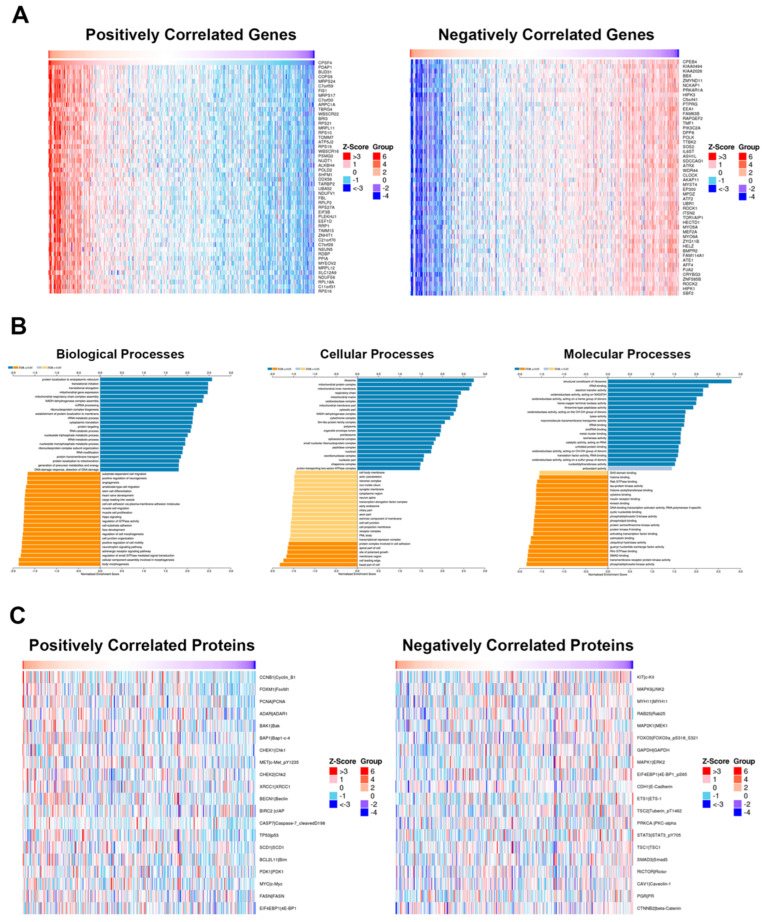
CPSF4 gene set enrichment study from prostate cancer patients’ TCGA-PRAD RNA-seq data. (**A**) Heatmap analysis showing the top 50 associated genes for each of the two groups of most up- and down-regulated genes, a total of 20,051 genes were analyzed. (**B**) Gene ontology (GO) enrichment analysis showing the biological, cellular, and molecular processes that are most positively and negatively associated, respectively. FDR values and Pearson correlation coefficients were used to rank the data. Using the Benjamini–Hochberg approach, the FDR is computed. (**C**) Heatmap linking the most up- and down-regulated proteins to the RNA expression of CPSF4 from patients in TCGA database.

**Figure 4 ijms-24-12961-f004:**
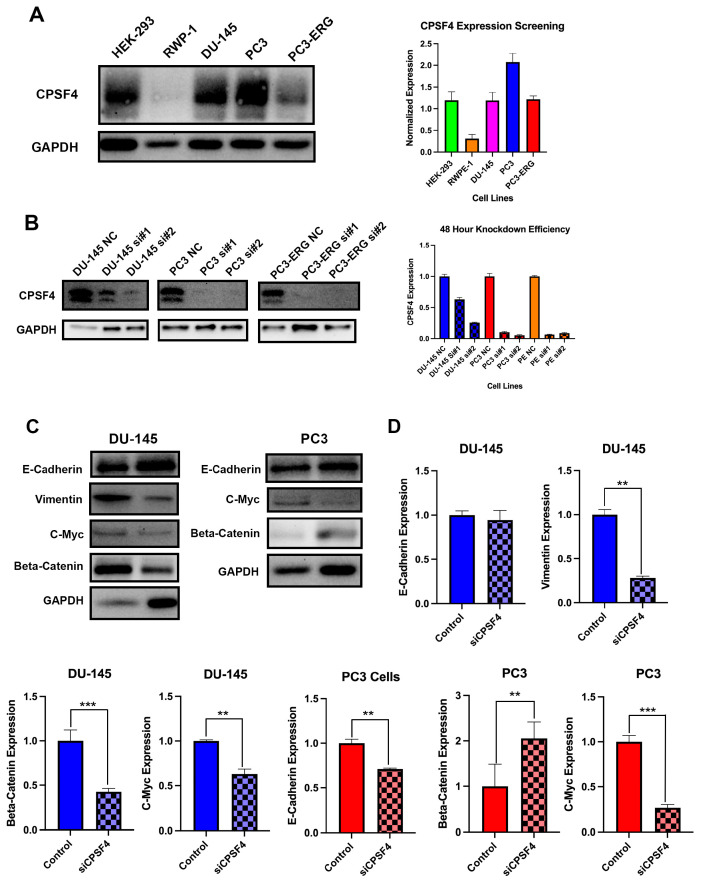
Analysis of CPSF4 using a PCa cell line Western blot. Blue graphs depict DU-145 cells and red graphs depict PC3 cells. (**A**) Western blot of the CPSF4 expression in the HEK-293, RWP-1, DU-145, PC3, and PC3-ERG cell lines, which are Pca cell lines. (**B**) After 48 h, the effectiveness of CPFS4 knockdown utilizing CPSF4 siRNA #1, #2, and scrambled siRNA as a negative control. (**C**) Western blotting study of the expression of the proteins E-Cadherin, Vimentin, c-Myc, and β-catenin in CPSF4 knockdown PC3 and DU-145 cells in comparison to control cells. GAPDH was employed as an internal quality control measure. (**D**) Graphical representation of c-Myc, β-catenin, ECAD, and Vimentin expression in siCPSF4 and control trials for DU-145 and PC3 cell lines (Asterisk ** *p<* 0.01 and *** indicates *p* < 0.001).

**Figure 5 ijms-24-12961-f005:**
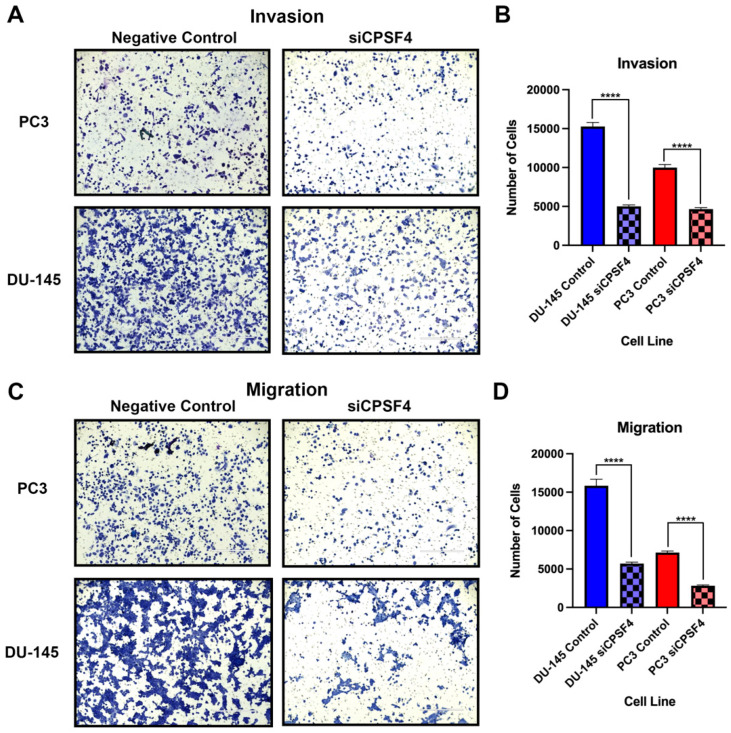
Prostate cancer cells are less able to move and invade when CPSF4 is inhibited. (**A**) Data of the invasion assay for the PC3 and DU-145 cell lines treated with siCPSF4, and scramble RNA (negative control) are shown. The scale bar indicates 400 μm, and the data were obtained from 3 repetitions. (**B**) Quantification of the invasive cell population from the PC3 and DU-145 cells invasion assay. (**C**) Images of the migration assay findings for the PC3 and DU-145 cell lines from three duplicate groups, including both control and CPSF4 knockdown treatment groups. (**D**) A graphic depiction of PC3 and DU-145 cell migration numbers in control and siCPSF4-treated cells. Blue graphs depict DU-145 cells and red graphs depict PC3 cells. (Asterisk **** indicates *p* < 0.0001). Scale bar in the figure represents 400 μm.

**Figure 6 ijms-24-12961-f006:**
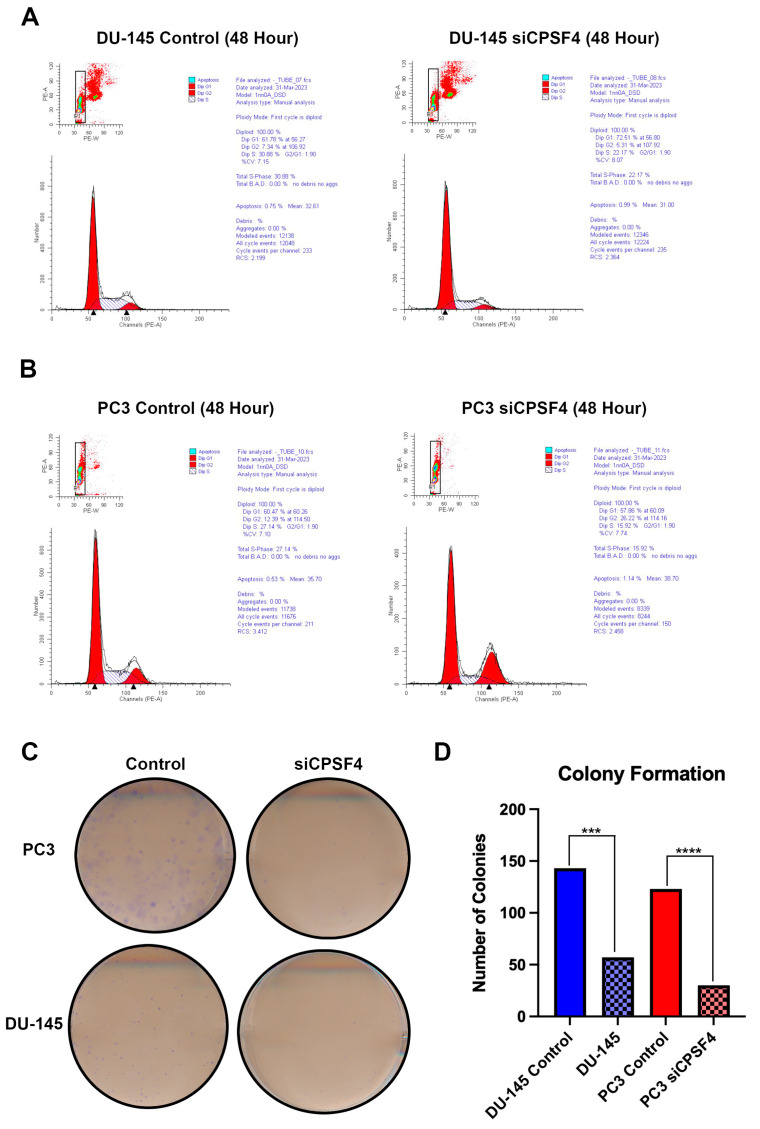
Dysregulation of cell cycle progression and signaling results from CPSF4 knockdown. (**A**) Analysis of the cell cycle and apoptosis using three replicates of flow cytometry on DU-145 control and siCPSF4-treated cells. (**B**) Analysis of the cell cycle and apoptosis in the siCPSF4-treated PC3 group versus the untreated control group. (**C**) Pictures of PC3 and DU-145 cell lines treated with siCPSF4 and scramble siRNA (negative control) from a colony formation test. (**D**) A graphic representation of a quantified analysis of a colony formation experiment (Astericks *** indicates *p* < 0.001 and **** indicates *p* < 0.0001). Dark column represents control trial and lighter/checkered column shows knockdown trial.

## Data Availability

Not applicable.

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
