# Peer review of "Cleavage and Polyadenylation-Specific Factor 4 (CPSF4) Expression Is Associated with Enhanced Prostate Cancer Cell Migration and Cell Cycle Dysregulation, In Vitro"

_ijms, 2023, doi:10.3390/ijms241612961_

Round 1

Reviewer 1 Report

This manuscript by Choudhry et al. aims to use TMA, TCGA and cell line data to explore the role of CPSF4 in the context of prostate cancer patients. 

1. The first paragraph in the result section (comparison of CPSF4 expression) is not clearly written. Please consider rewording. 

2. In Figure1, what is included in Gleason Group1 and Group2? Maybe the authors should just use the actual Gleason scores to describe the CPSF4 expression association. And how did the authors conclude that the expression of CPSF4 is associated with PCa onset? 

3. What is cancer and what is benign in Figure2? And in Figure2, what test did the author use to compare TPM expression between different groups? 

4. For the correlation analysis, did the authors also filter the genes by the TPM expression? Because it is possible that a given gene is significantly correlated with CPSF4 but it is lowly expressed. Therefore implementing a threshold of expression is essential. And why did the authors only pick the top50 genes? How many genes are included in the GO analysis? Please specify. And please also increase the resolution. The GO analysis results are impossible to read.

Author Response

Please see the attachment."

Reviewer 2 Report

Overall, the manuscript focuses on the CPSF4 functions in prostate cancer. CPSF4 is highly expressed in prostate cancer in their unique 353 patients cohort.CPSF4 could promote cancer progression by regulating different cell behavior like the cell migration cell cycle arrest et al.

Some comments on this manuscript:

Please state explicitly in the manuscript that all patients who participated in this study provided written informed consent for the collection and research use of their materials. Additionally, mention that the use of these samples was approved by the appropriate ethics committee or institutional review board.

Regarding the TMA/IHC cohort, it would be valuable to include information about the prognosis effect of CPSF4 in prostate cancer. Consider discussing any correlations between CPSF4 expression levels and patient outcomes, such as overall survival or disease-free survival.

Increase the resolution of all figures to ensure better clarity and readability for the readers.

In Figure 3C, please specify the origin or source of the protein dataset used for analysis. Mention if it was from patient samples, cell lines, or any other source.

Address the issue of the two bands in the western blot for CPSF4. Clearly indicate which band corresponds to CPSF4 in the figure and provide a brief explanation for the presence of two bands, if known.

Provide the specific sequences of the siRNA used for knocking down CPSF4 and the primers used for quantitative PCR (qPCR) experiments. This information is essential for reproducibility.

In Figure 4D, label the significance levels or p-values to indicate which comparisons are statistically significant. This will help readers interpret the results more effectively.

Consider investigating the effect of knocking down CPSF4 expression on cell growth. This information would provide valuable insights into the functional role of CPSF4 in prostate cancer progression.

In Figure 6A, perform statistical analysis of the cell numbers in different cell cycle phases before and after knocking down CPSF4 expression. This will help demonstrate whether CPSF4 knockdown has any significant effect on cell cycle distribution.

Label the significance levels or p-values in Figure 6D to indicate the statistical significance of the results. This will help readers understand the significance of the observed differences.

To further validate the in vitro findings, consider conducting an in vivo study using animal models. This will help confirm the relevance and translational potential of the results.

Thoroughly proofread the manuscript to correct any minor grammar errors or typos that may have been overlooked during the initial preparation. Clear and concise language is crucial for effective communication of scientific findings

NONE

Reviewer 3 Report

The article by Muhammad Choudhry and co-authors is made in the field of modern approaches to the diagnosis and therapy of prostate cancer and is devoted to demonstrating that CPSF4 may be used as a possible biomarker in prostate cancer. The work is preceded by a fairly voluminous and detailed introduction summarizing the available literature data on prostate cancer in general, the molecular mechanisms of its development, as well as some information about Polyadenylation Specificity Factor Subunit 4 and its association with the development of prostate cancer. The works of recent years are presented the coverage of the available literature data on the discussed problem, mainly of the last decade. The main text of the article describes the study of the expression of the CPSF4 protein in a TMAs cohort which containing various prostate cancer stages. it was demonstrated in our cohort that higher levels of CPSF4 protein expression was associated with aggressive PCa subtypes. This article, in my opinion, may be useful to specialists in the field of biology and medicinal chemistry, engaged in research in the field of diagnosis and therapy of prostate cancer, and can be recommended for publication after making the following minor adjustments. In general, I have no comments on the results, but there are some shortcomings in their presentation and discussion.   1. It is recommended to expand the list of references in the Introduction; after all, 11 references in this section are not enough for a journal of a high scientific level. 2. The discussion of the results is also insufficient and needs to be expanded. 3. In the Conclusions part, it is undesirable to quote the phrase "Further studies are needed to document the mechanisms underpinning CPSF4 effects on the proliferation pathways in PCa and other forms of cancer", perhaps it would be better to make some assumptions about this and give them in the Discussion section.

Minor editing of English language may be required

Round 2

Reviewer 1 Report

The authors have adequately addressed all the previous comments. 

Author Response

We would like to thank you for reading and reviewing our article. Your considerate observations 
and thorough criticism were really helpful in forming the paper's final form. Your 
meticulousness and the helpful comments you offered unquestionably improved the quality of 
the work. Your generosity in sharing your knowledge and providing constructive criticism has 
been really beneficial to me. Your commitment to the peer review process is admirable, and I 
sincerely appreciate what you have done to improve this field of study. I appreciate your 
assistance and time. 

Reviewer 2 Report

NONE

Author Response

We would like to express our gratitude for your kindness in reading and reviewing our article. 
The ideas offered have been significantly improved by your insightful analysis and feedback. 
Your comments not only pointed out places that needed improvement, but also provided novel 
viewpoints that significantly improved the content. Your dedication to advancing intellectual 
conversation is demonstrated by your desire to contribute your time and knowledge. I sincerely
appreciate your contribution to the production of this book, and I am grateful for your 
generosity in sharing your ideas and knowledge. We appreciate your significant help.